# Modification of Hand Muscular Synergies in Stroke Patients After Robot-Aided Rehabilitation

Francesco Scotto di Luzio [1,*,†] , Francesca Cordella [1] , Marco Bravi [2] , Fabio Santacaterina [2] , Federica Bressi [2] , Silvia Sterzi [2] and Loredana Zollo [1]

1   Research Unit of Advanced Robotics and Human-Centred Technologies, Università Campus Bio-Medico di Roma, Via Alvaro del Portillo, 21, 00128 Rome, Italy; f.cordella@unicampus.it (F.C.); l.zollo@unicampus.it (L.Z.)

2   Research Unit of Physical Medicine and Rehabilitation, Università Campus Bio-Medico di Roma, Via Alvaro del Portillo, 21, 00128 Rome, Italy; m.bravi@policlinicocampus.it (M.B.); f.santacaterina@policlinicocampus.it (F.S.); f.bressi@policlinicocampus.it (F.B.); s.sterzi@policlinicocampus.it (S.S.)

*   Correspondence: f.scottodiluzio@unicampus.it

**Abstract:** The central nervous system (CNS) is able to control a very high number of degrees of freedom to perform complex movements of both upper and lower limbs. However, what strategies the CNS adopts to perform complex tasks are not completely clear and are still being studied. Recent studies confirm that stroke subjects with mild and moderate impairment show altered upper limb muscle patterns, but the muscular patterns of the hand have not completely investigated, although the hand represents a paramount tool for performing activities of daily living (ADLs) and stroke can significantly alter the mobilization of this part of the body. In this context, this study aims at investigating hand muscular synergies in chronic stroke patients and evaluating some possible benefits in the robot-aided rehabilitation treatment of the hand in these subjects. Seven chronic stroke patients with mild-to-moderate impairment ($FM > 30$) were involved in this study. They received a 5-week robot-aided rehabilitation treatment with the Gloreha hand exoskeleton, and muscle synergies of both the healthy and injured hand were evaluated at the beginning and at the end of the treatment. The performed analysis showed a very high degree of similarity of the involved synergies between the healthy and the injured limb both before and after the rehabilitation treatment (mean similarity index values: H-BR: $0.88 \pm 0.03$, H-AR: $0.94 \pm 0.03$, BR-AR: $0.89 \pm 0.05$). The increasing similarity is regarded as an effect of the robot-aided rehabilitation treatment and future activities will be performed to increase the population involved in the study.

**Keywords:** muscular synergies; robot-aided rehabilitation; non-negative matrix factorization algorithm; stroke; hand

## 1. Introduction

Motor coordination represents one of the most fascinating aspects of human nature. The central nervous system (CNS) is able to control a very high number of degrees of freedom to perform complex movements of both upper and lower limbs [1]. However, the strategies that the CNS uses to perform complex tasks are not yet completely clear and are still being studied [2]. Many studies have shown the existence of motor primitives, which would allow the CNS to manage the complex architecture of the human body, guaranteeing complex movements starting from simple movements [3]. Muscular synergy, from the Greek language, means "working together" and was introduced for the first time in a work of Bernstein as "*. . . solution to the problem of selecting one movement among the infinite possibilities of motor solutions to perform a specific task. . .*" [4,5]. It has been suggested that complex movements are constructed through smaller blocks (i.e., muscular synergies) able to involve different muscle groups, in order to overcome the difficulties related to the coordination of a high number of degrees of freedom [1,3,6].

The upper limb and the hand are very complex districts of our body, able to perform very precise movements, such as grasping and manipulating of small objects [2]. A detailed understanding of the mechanisms governing such complex movements would have enormous implications in the rehabilitation and prosthetic fields.

Muscle synergies are extracted from electromyographic (EMG) signals acquired from muscles involved in the movement. Several algorithms can be adopted to extract muscle synergies from EMG signals, such as non-negative matrix factorization (NNMF) [7], principal component analysis (PCA), factory analysis (FA), and inverse Gaussian [8–10]. The goal of such algorithms is to reduce the number of variables describing the EMG dataset, limiting the loss of information as much as possible [10].

The first studies concerning muscle synergies aimed at demonstrating that motor control could be described by a set of muscle synergies. Several studies have been conducted to select the number of synergies to be extracted and to evaluate the stability of synergies. The results already presented in the state-of-the-art have been very promising and have pushed researchers to investigate the role of muscular synergies in different fields of application ranging from sport to clinic and robotics [9]. Unfortunately, there is not a defined protocol on how many and which muscles should be selected to analyze a certain movement or a certain synergy. This choice is left to the experience of the researcher who selects the muscles on the basis of the task to be performed [3].

The monitoring of muscle patterns would allow having a quantitative picture of patients suffering from neurological disorders, such as stroke and musculoskeletal pathologies, in general for both upper and lower limbs [11–14]. An advantage of this analysis is the unobtrusiveness of the adopted sensors and the ease of execution, since the analysis and extraction of muscle synergies can be performed offline also with superficial EMG sensors. Several studies have been conducted on the alteration of the muscular synergies in subjects affected by stroke. In particular, the anomalies of the synergies of the upper limb in subjects with strong and moderate impairment were analyzed. The results showed a similarity between the synergies of the healthy and the affected limb [15]. From the literature analysis, it is evident that the presence of a particularly strong coupling of elbow flexion and shoulder movements in stroke patients affects reaching movements [15,16]. The deficit in motor performance of patients affected by stroke is due to alterations in the activation of muscular patterns. Furthermore, patients with significant neurological damage often have alterations in motor performance, evident from the analysis of muscle synergies, as observed in [17–19]. After a stroke, the human brain puts in place a reorganization of healthy tissue, favoring a progressive recovery of functions of the injured part of the CNS. This neural plasticity is more evident in the acute and sub-acute phase and tends to decrease in the chronic phase, as described in [20]. Over the years, the increased incidence of this pathology has pushed researchers to investigate rehabilitation solutions able to provide an adequate level of recovery and the restoration of motor function.

The introduction of robotic platforms for rehabilitation should favor the progressive recovery of muscle patterns. However, it is necessary to ensure that the use of the robot does not introduce disturbing factors in the muscle patterns [21]. Few studies have been devoted to exploring the impact of stroke in motor control, to analyze how, and how much, robot-aided rehabilitation allows improving hand control and manipulation [15,18,22], also in combination with conventional treatment or other, such as transcranial direct current stimulation [23]. Different outcomes of robot-aided rehabilitation treatment were observed, on the basis of the time elapsed since neurological damage. For acute patients, researchers observed a reduction in the number of synergies compared to the number of synergies extracted from the healthy limb [24]. Moreover, patients in the acute phase show a reduction in the number of synergies on the healthy side before a robot-aided rehabilitation treatment. This result should be interpreted, taking into account that this category of patients demonstrates a tendency for splitting and merging of muscle synergies [15,25]. In chronic stroke patients, the structure of muscle synergies involved in movement appears to remain unaltered, but the modulation of such synergies is often compromised [15,26].



Furthermore, the patient with impaired motor functions tends to show an adaptation of muscle synergies to typical characteristics of movement (such as type of movement, speed, compensation of asymmetry due to neurological damage) [15,25,26].

In such a context, the human hand represents one of the most complex anatomical districts of the human body and, thanks to more than 20 degrees of freedom, allows performing gripping and manipulation tasks. Some studies have shown that muscle synergies represent a tool for a complete description of grip control and that they can be adopted as a predictive tool to generate new hand postures [2]. An accurate analysis of muscular and postural synergies was carried out in [27]. This study showed that healthy subjects who performed gripping tasks in different configurations show a statistical overlapping among muscle synergies. This result is not trivial, as it allows us to demonstrate not only the existence of muscle synergies, but also the role they play in terms of control of the different grips [28]. Moreover, recent studies confirm that stroke subjects with mild and moderate impairment show altered upper limb muscle patterns when compared with healthy subjects during the performance of hand-reaching tasks in different directions of the space [29].

To the best of our knowledge, there are no studies analyzing the motor patterns of the hand of subjects affected by stroke after a robot-aided rehabilitation treatment, although it represents a tool of fundamental importance for performing activities of daily living (ADLs), and stroke can significantly alter the mobilization of this part of the body. Muscle synergies are useful to recognize alterations during the execution of motor tasks, since they allow highlighting the contribution of the single components that constitute complex movements in chronic stroke patients [3]. Muscular synergies of the upper limb have been adopted to evaluate possible benefits of robot-aided rehabilitation in post-stroke patients. Furthermore, it has been demonstrated a similarity between muscle synergies of affected and unaffected limbs. Similar studies have not been performed on the hand, yet. For these reasons, the aim of this work is to investigate hand muscular synergies of chronic stroke patients before and after rehabilitation treatment performed with the Gloreha Sinfonia exoskeleton. It is evident that the studies carried out so far on the muscular synergies of the hand have not shown the same evidence obtained for the upper limb in stroke subjects in combination with robot-aided rehabilitation, in order to evaluate its possible benefits on improving hand dexterity and bring muscle synergies closer to those of the unaffected limb. The advancement, compared to the state-of-the-art, is twofold: (i) to investigate the muscular synergies of the hand in subjects affected by stroke, and (ii) to quantify the effects of the robot-aided rehabilitation treatment of the hand in these subjects.

The paper is organized as follows: in Section 2, the analysis of muscular synergies, in combination with robot-aided rehabilitation, and the experimental protocol are described. Sections 3 and 4 are focused on the experimental results obtained with post-stroke patients and their discussion. Conclusions and future work are reported in Section 5.

## 2. Materials and Methods

### 2.1. Subjects

Seven chronic stroke patients (mean age: $59.6 \pm 12.8$) with mild and moderate impairment were involved in this study, as reported in Table 1. Chronic patients, although characterized by reduced level of plasticity and recovery rate with respect to acute or sub-acute patients, were involved in this study since an adequate level of muscle activation was needed to extract muscle synergies. This level, estimated by adopting inclusion criteria based on the clinical Fugl–Meyer and motor power scales, was reached only by chronic patients. The subjects had to perform treatment with the Gloreha Sinfonia (IDROGENET, Brescia, Italy) [30], a robotic glove for hand rehabilitation. At the beginning and the end of the course of treatment, the subjects' electromyographic (EMG) signals were recorded in order to compute hand muscular synergies.

**Table 1.** Summary table of patients involved in this study.

|   | Age | Sex | Stroke | Time from Stroke [Months] |
|---|-----|-----|--------|---------------------------|
| 1 | 56 | M | ischemic | 68 |
| 2 | 53 | F | hemorrhagic | 96 |
| 3 | 40 | M | ischemic | 75 |
| 4 | 79 | F | ischemic | 18 |
| 5 | 61 | M | hemorrhagic | 49 |
| 6 | 56 | M | hemorrhagic | 30 |
| 7 | 72 | M | ischemic | 33 |
|   | $59.6 \pm 12.8$ |  |  | $52.7 \pm 28.1$ |

### 2.2. Experimental Protocol

As reported in Figure 1, each patient carried out a robot-aided rehabilitation treatment with the Gloreha Sinfonia, composed of five sessions per week for 4 weeks. At the beginning and at the end of the course of treatment with robot (i.e., before rehabilitation and after rehabilitation, BR and AR, respectively), sEMG signals were recorded from injured and healthy hand, by using the Delsys Trigno EMG wireless system, in order to extract muscular synergies of each subject and evaluate patient rehabilitation outcome. Moreover, each subject was evaluated using Fugl–Meyer motor assessment of the upper extremity (FM) and motor power (MP) assessment before and after the therapy program with the Gloreha robot [31,32].

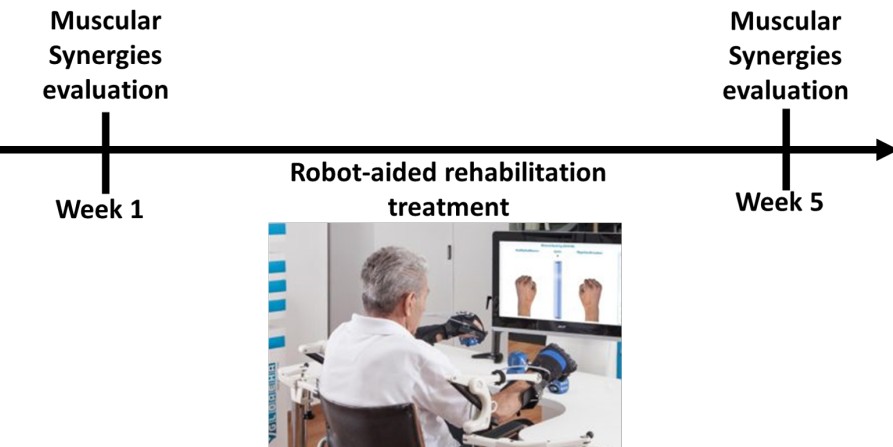

**Figure 1.** An overall view of the experimental protocol.

As showed in Figure 2, the recording of the EMG signals was performed with the patient sitting in a comfortable position, in front of a screen that showed him/her the grip to perform. The subject performed four different grasps selected randomly, shown by means of a purposely developed user interface and performed similarly to the FM assessment. He/she had to grab and hold the following objects: a can, a pencil, a sheet, and a tennis ball (Figure 3). The subjects were required to perform the grasp and to hold the object for ten seconds, followed by a pause of at least ten seconds. The sEMG data were recorded during the ten seconds of grasp execution. The subjects were asked to perform the same task with the injured limb before and after rehabilitation treatment (BR and AR) and subsequently with the healthy limb in order to obtain the healthy muscle patterns (H), i.e., a comparison term for each subject. Furthermore, at the beginning and at the end of the treatment, the subjects were assessed by means of the FM and MP scales to evaluate the performance through clinical validated tools.

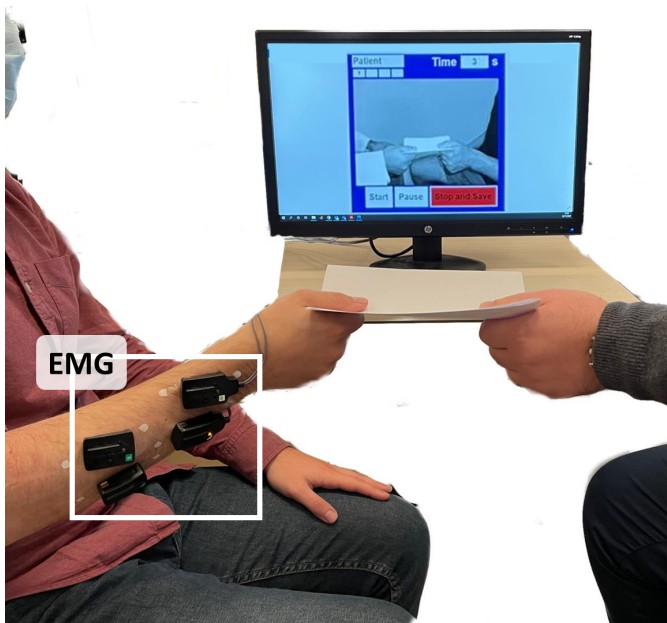

**Figure 2.** The experimental protocol for muscular synergies evaluation at the admission and discharge of the patient.

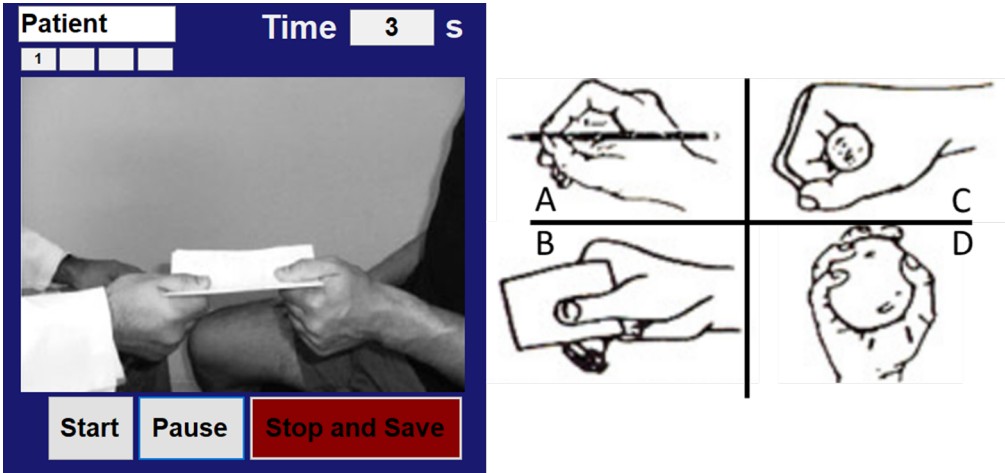

**Figure 3.** The adopted user interface is reported on the left. The therapist can use the buttons in the bottom to start the timer, pause it, and to stop and save the data. The tasks performed by the patient are reported on the right: grasp a pencil (**A**), grasp a sheet (**B**), grasp a small can (**C**), and grasp a tennis ball (**D**).

The study was conducted under Ethical Committee approval (Ethical Approval N. 41/17 OSS Com Et CBM) and in accordance with the Declaration of Helsinki. All patients were adequately informed about the purpose of the study and gave their written informed consent.

### 2.3. Gloreha Sinfonia

The Gloreha Sinfonia (Figure 4) is a robotic exoskeleton for robot-aided neuro-rehabilitation of the hand. It is composed of a motorized glove for hand mobilization driven by five permanent magnetic actuators (LA12 Actuator, TECHLINE). The cable actuation of the glove allows placing motors distant from the patient and to administer the therapy in total safety for him/her. The motors are, in fact, positioned in a mobile box. Furthermore, the Gloreha Sinfonia is equipped with a second glove that is not actuated, but equipped with five bend-sensors (Flexpoint Inc., West Jordan, UT, USA) to

detect the movement of the healthy limb (i.e., of the corresponding finger) and replicate it on the injured limb. Both gloves are connected to the PC via USB cable, and a graphic user interface for carrying out functional tasks during the rehabilitation treatment is provided. This software implements different kind of tasks, such as grasping and moving objects, and records data about performance of the patient during robot-aided rehabilitation. Two passive arm supports are used to treat patients who cannot support the weight of their limb. The Gloreha Sinfonia allows carrying out both one-handed or bimanual tasks. In this study, the Gloreha hand exoskeleton was used in active-assisted mode during the rehabilitation treatment, involving both limbs of the patient. Indeed, the patient was asked to independently start the motor task with the sound hand and performed movements to execute simple grasping tasks, maximizing his/her involvement. The robotic glove follows the patient's activity and the motorized system intervenes to help, if necessary. Thus, the patient's involvement and motivation are amplified thanks to the use of both hands in the execution of functional rehabilitation tasks.

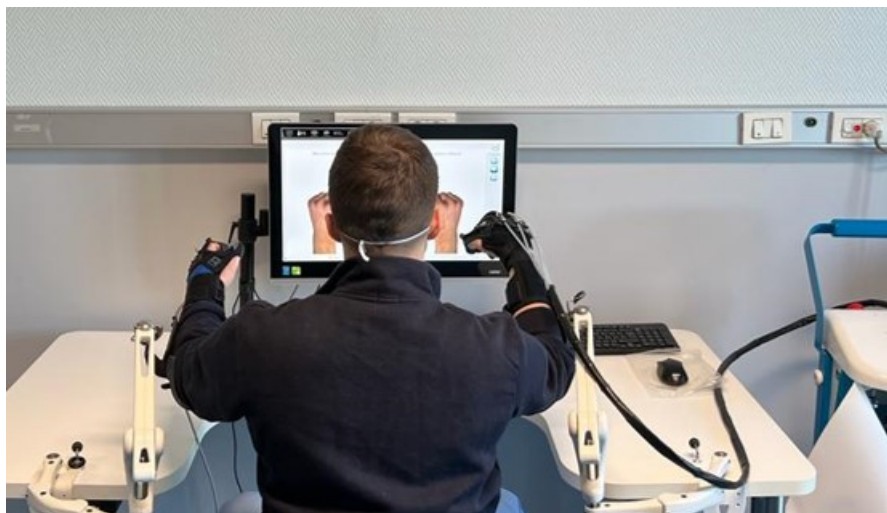

**Figure 4.** The Gloreha Sinfonia, the robotic exoskeleton for the neuro-rehabilitation of the hand.

### 2.4. EMG Acquisition and Muscle Synergies Extraction

SEMG signals were acquired from six muscles of the hand: flexor digitorum superficialis (FDS), exstensor digitorum superficialis (EDS), flexor pollicis brevis (FPB), abductor pollicis brevis (APB), abductor digiti minimi (ADM), and exstensor digiti minimi (EDM). Before placing the electrodes, the hairs were removed from the skin surface and cleaned thoroughly with alcohol. The sEMG data were collected at 1 kHz and stored in a text file for offline analysis. A visual inspection was performed to avoid the presence of artifacts that contaminated the signal. They were filtered with a sixth-order Butterworth filter with cutoff frequencies of 30,450 Hz and a second-order Butterworth notch filter (50 Hz) and then rectified to extract muscle synergies. Muscle synergies were extracted from EMG signals by using the non-negative matrix factorization (NNMF) algorithm [33] as

$$E(t) = \sum_{i=1}^{n} (W_i * H_i(t)) + e \tag{1}$$

where $E(t)$ is an N × M EMG signals matrix (N muscles and M number of the samples), W is the N × S synergy matrix (S number of synergies), and H is the T × M coefficient matrix. Each column of W contains the weights of each muscle for the corresponding synergy, and each row of H represents how much the corresponding synergy was activated or used to generate force. With this assumption, the contribution of a single muscle to the performed task is given by the linear combination of the product between the muscle gain $W_i$ and the corresponding activation coefficient $H_i(t)$.

For muscle synergies extraction, the NMF algorithm was initialized with random matrices with elements belonging to a uniform distribution with values between 0 and 1, then updated until convergence, so that the coefficient of determination $R^2 < 0.01\%$. The synergy extractions were iterated 50 times for each limb of each subject and the synergies with the highest $R^2$ were selected for further comparisons, because the obtained solutions for the synergies and their coefficients found by the algorithm may represent a local extremum on the $R^2$ surface ([15]). For this reason, the largest $R^2$ was selected for subsequent analysis, in order to ensure positioning as close as possible to the absolute maximum on the surface of the $R^2$. The value of 50 was chosen as it is considered a good trade-off between the reconstruction of the EMG signal, evaluated with the $R^2$, and the computational burden for the iterated calculation. The number of extracted synergies was chosen on the basis of $R^2$. In other words, the EMG factorization is performed in order to minimize the difference,

$$e(t) = E(t) - W_i * H_i(t), \tag{2}$$

between the acquired EMG signal and the reconstructed one. The number of synergies N extracted from the EMG signal is the minimum value that returns $R^2$ at least equal to 80% ([15]). More in detail, in order to select the minimum number of synergies N, suitable for a complete representation of the recorded EMG signal $E(t)$, the sets of muscle synergies were extracted for each subject with N ranging from 1 to 6 in order to identify the minimum number of synergies for which the condition $R^2 > 80\%$ is satisfied. At the end of this analysis, the mean value of synergies N on all the patients was considered, in order to obtain comparable synergy sets.

### 2.5. Performance Indices

The muscular synergies extracted during the assigned tasks in the three conditions (i.e., H, BR, and AR) were compared for the same patient using two similarity indices:

- Cosine similarity (CS): it is a measure of the degree of similarity between two different vectors, described by Equation (3). It is the measure of the cosine of the angle between the two non-zero vectors $W_i$ and $W_j$, and is a value ranging from $-1$ to 1. The CS is equal to 0, when the angle measured between the two vectors is equal to $\pi/2$. In our analysis, a CS close to 1 indicates equal synergies, and values close to $-1$ indicate vectors of equal but opposite synergies. It can be computed as

$$CS = cos(\theta) = \frac{W_i * W_j}{\| W_i \| \| W_j \|} \tag{3}$$

  where $\theta$ is the angle between the two synergy vectors $W_i$ and $W_j$, $\| W_i \|$ is the norm of the synergy vector, and $*$ denotes the scalar product.

- Similarity index (SI): it is computed as the weighted sum of the difference between two synergy vectors, as described by Equation (4). SI represents a similarity index between two synergy vectors, able to take into account the Euclidean distance between the two vectors of synergies. Hence, two vectors with a high SI are in the same region of the vector space. It can be computed as

$$SI = 1 - \frac{1}{S} \sum_{i=1}^{n} |W_i - W_j| \tag{4}$$

  where $S$ is the number of muscular synergies, and $W_i$ and $W_j$ are the two synergy vectors.

### 2.6. Statistical Analysis

A statistical analysis based on Wilcoxon paired-sample test was conducted in order to analyze the results obtained by the FM and MP scales (the significance was achieved for $p < 0.05$) and to evaluate the statistical significance between the muscular synergy sets extracted in the three considered conditions (i.e., H, BR, and AR) and for the two

performance indices calculated (CS and SI). In this case, the significance was achieved for $p < 0.02$ (with Bonferroni correction).

## 3. Results

Data about the clinical evaluation acquired at admission and discharge of the patients are reported in Table 2. All the patients showed an increase in the FM and MP scores between admission and discharge with the Gloreha Sinfonia. The average increase was $7.3 \pm 3.7$ for the FM and $3.1 \pm 2.1$ for the MP. This underlines the improvement in the mobilization of the injured limb in the selected population, and this improvement is statistically significant (single comparison, significance achieved for $p < 0.05$; in this case, $p = 0.02$ for FM and $p = 0.03$ for MP with Wilcoxon paired-sample test).

**Table 2.** Clinical data of patients involved in the study. BR = before rehabilitation; AR = after rehabilitation; * denotes statistical significance.

| | FM * | | MP * | |
|---|---|---|---|---|
| | **BR** | **AR** | **BR** | **AR** |
| 1 | 39 | 45 | 5 | 10 |
| 2 | 37 | 48 | 9 | 12 |
| 3 | 36 | 48 | 6 | 12 |
| 4 | 45 | 51 | 15 | 18 |
| 5 | 34 | 37 | 4 | 8 |
| 6 | 31 | 34 | 16 | 16 |
| 7 | 26 | 36 | 14 | 15 |
| | $35.4 \pm 6.0$ | $42.7 \pm 6.9$ | $9.9 \pm 5.1$ | $13.0 \pm 3.5$ |

Figure 5 shows the trend of $R^2$ for the selected tasks as the number of synergies increases. In particular, the $R^2$ between the EMG signals and the reconstructed ones $(W * H)$ is shown for the healthy limb and the injured limb before and after treatment with the Gloreha robot. It is evident that for $N = 3$, the $R^2 > 80\%$, therefore three synergies were extracted from the EMG signal in order to allow the comparisons between the different conditions examined (i.e., H, BR, and AR).

Muscle synergies were extracted and compared with the recorded EMG data of the 7 muscles selected at the beginning and at the end of the treatment on each subject and on the contralateral limb. Since the synergies are subject-specific, multiple comparisons were made between the beginning and the end of the rehabilitation treatment of the affected and the sound hands. For the sake of brevity, the results concerning contributions of each muscle synergy are shown, for a representative subject, in Figures 6–9. Similar results were obtained for the other involved subjects. In particular, the first bar is the rate of the corresponding muscle in the healthy limb (H), and the second and the third bars are the rate of each muscle before and after rehabilitation (BR and AR), respectively.

The results concerning the performance indices for each exercise are summarized in Table 3 for all the involved patients. The monitored patients show a very high degree of similarity of the involved synergies (mean CS values: H-BR: $0.74 \pm 0.09$; H-AR: $0.91 \pm 0.06$; BR-AR: $0.82 \pm 0.09$).

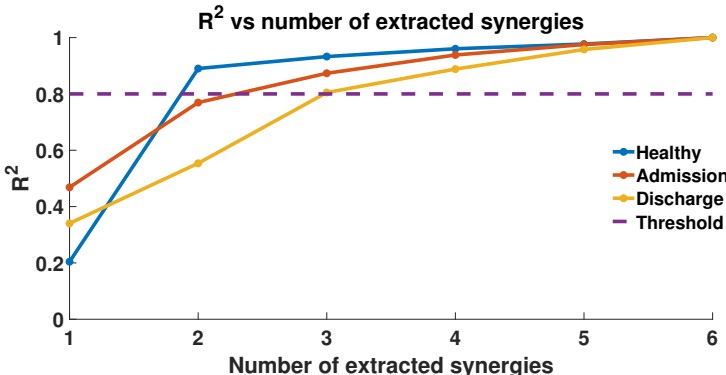

**Figure 5.** The coefficient of determination $R^2$ computed between mean EMG data and reconstructed EMG data for healthy limb and affected one at the admission and discharge. For N = 3, $R^2$ is above the threshold (0.8) in all the assessed conditions.

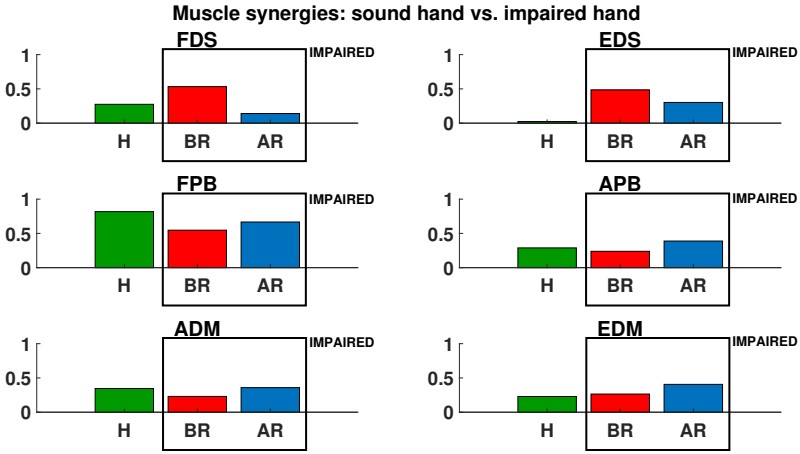

**Figure 6.** Comparison of the muscular synergies of the sound hand (in green) vs. the impaired hand (in the box) before (red) and after (blue) the rehabilitation treatment with the Gloreha robot for pencil-grasping.

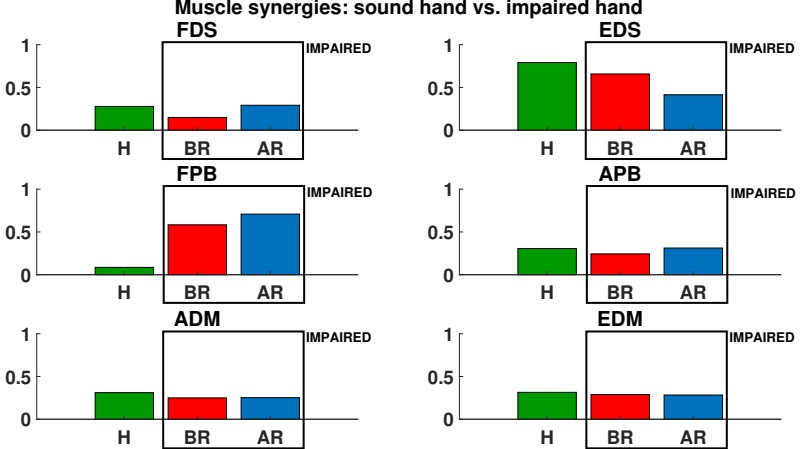

**Figure 7.** Comparison of the muscular synergies of the sound hand (in green) vs. the impaired hand (in the box) before (red) and after (blue) the rehabilitation treatment with the Gloreha robot for sheet-grasping.

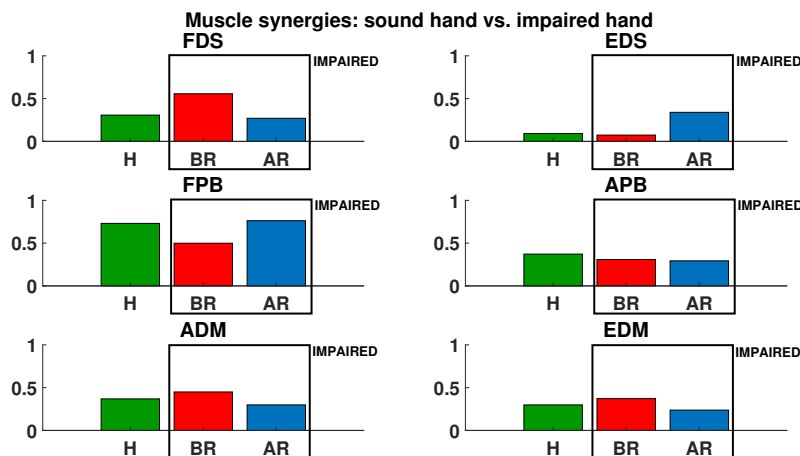

**Figure 8.** Comparison of the muscular synergies of the sound hand (in green) vs. the impaired hand (in the box) before (red) and after (blue) the rehabilitation treatment with the Gloreha robot for can-grasping.

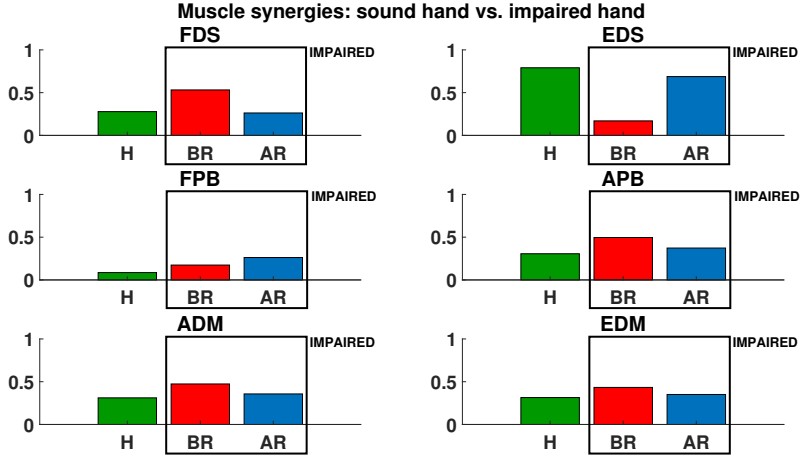

**Figure 9.** Comparison of the muscular synergies of the sound hand (in green) vs. the impaired hand (in the box) before (red) and after (blue) the rehabilitation treatment with the Gloreha robot for ball-grasping.

**Table 3.** Summary table of comparisons among muscular synergy vectors for each exercise. P = pencil; S = sheet; C = cylinder; B = ball; * denotes statistical significance.

|  | CS | | | SI | | |
|---|---|---|---|---|---|---|
|  | **H-BR *** | **H-AR *** | **BR-AR *** | **H-BR *** | **H-AR *** | **BR-AR *** |
| P | $0.87 \pm 0.08$ | $0.94 \pm 0.03$ | $0.92 \pm 0.06$ | $0.91 \pm 0.04$ | $0.95 \pm 0.05$ | $0.92 \pm 0.04$ |
| S | $0.75 \pm 0.13$ | $0.85 \pm 0.05$ | $0.87 \pm 0.10$ | $0.86 \pm 0.03$ | $0.94 \pm 0.03$ | $0.84 \pm 0.02$ |
| C | $0.70 \pm 0.22$ | $0.98 \pm 0.01$ | $0.72 \pm 0.19$ | $0.85 \pm 0.04$ | $0.90 \pm 0.02$ | $0.85 \pm 0.03$ |
| B | $0.65 \pm 0.14$ | $0.87 \pm 0.09$ | $0.76 \pm 0.20$ | $0.89 \pm 0.05$ | $0.96 \pm 0.04$ | $0.93 \pm 0.04$ |
|  | $0.74 \pm 0.09$ | $0.91 \pm 0.06$ | $0.82 \pm 0.09$ | $0.88 \pm 0.03$ | $0.94 \pm 0.03$ | $0.89 \pm 0.05$ |

## 4. Discussion

Seven chronic stroke patients (mean age: $59.6 \pm 12.8$) were involved in this study. All the involved patients showed an increase in the FM and MP scores between before and after the robot-aided rehabilitation of the hand. The average increase was $7.3 \pm 3.7$ for the FM and $3.1 \pm 2.1$ for the MP.

Moreover, a similarity among the synergy patterns extracted from the EMG signals is already evident from a visual inspection of the results reported for the representative subject. The monitored patients show a high degree of similarity of the involved synergies, as evident from the comparison reported in Table 3. This also manifests in the comparison between the healthy limb and the injured limb before and after the rehabilitation treatment.

These results confirm what has been demonstrated in the state-of-the-art with regard to muscle synergies of the upper limb on chronic stroke patients. More in detail, in the state-of-the-art, a preservation of the normal muscle synergies in the stroke-affected arm of a patient, with a similarity level ranging from 0.89 to 0.98, has been demonstrated [15]. The obtained results aim to extend the results to the hand district and to the robot-mediated rehabilitation treatment district. The increase in CS following rehabilitation treatment can be synonymous with its effectiveness, even if a larger population would be needed. This aspect is also confirmed by the statistical analysis; in fact, the increase in CS is statistically significant in H-BR, H-AR, and BR-AR comparisons ($p$ = 0.018, 0.012, 0.015 with Wilcoxon paired-sample test, respectively).

The comparison BR-AR reveals a high mean value of CS ($0.82 \pm 0.09$) with two interesting aspects to be considered in terms of motor performance: (i) each patient tends to maintain the synergistic pattern as almost unchanged even after the rehabilitation treatment, although the clinical results show an improvement in motor performance; (ii) motor performance deficits in chronic stroke subjects are not strictly related to the contribution of the single muscle in movement, but to how these are modulated over time (i.e., the corresponding activation coefficient $H_i(t)$).

Furthermore, the results of the comparisons made with SI for each patient are reported in Table 3. The similarity between the vectors of synergies is good both for the comparisons between the healthy limb and the injured limb before and after the robot-aided rehabilitation (mean SI values: H-BR: $0.88 \pm 0.03$; H-AR: $0.94 \pm 0.03$; BR-AR: $0.89 \pm 0.05$). The high average SI level, also confirmed by the statistical analysis on the analyzed population ($p$ = 0.016, 0.015, 0.012 with Wilcoxon paired-sample test, respectively), suggests a high degree of similarity of the muscle pattern both before and after rehabilitation and in comparison with the contralateral limb.

All the involved patients show similar muscle synergies between the injured limb and the healthy limb, despite a motor deficit as evidenced also by the results of the clinical scales. The introduction of a robot-aided rehabilitation system can promote motor recovery, as evident from the comparisons. In order to obtain a complete overview of the mechanisms underlying motor coordination in healthy subjects and to investigate it on stroke patients with different levels of severity, it is necessary to extend the study to a higher number of muscles with a standardized protocol. These considerations have been obtained for patients with mild-to-moderate impairment ($FM \geq 30$, see Table 2).

This analysis also investigates the feasibility of using muscular synergies as a marker to identify anomalies in muscle activation patterns, not only on the shoulder–elbow district, as in [15], but also for the hand. This analysis allows describing a complete picture of the subject affected by stroke, comparing the data of the injured limb and the motor pattern extracted on the contralateral side of the same subject, making the evaluation is extremely accurate and custom-built on each patient.

The results obtained from the clinical scales show an improvement in terms of motor performance and coordination. These results, although encouraging, were obtained on seven stroke patients and should be investigated in more detail involving a wider population of chronic stroke patients in order to avoid the number of enrolled patients affecting the statistical significance. The obtained results have a 95% confidence interval with a confidence level of about 40%.

The robot-aided rehabilitation treatment determines a variation of the muscular pattern of the patient between admission and discharge. This result is paramount, because it demonstrates that the patient tends to remodel the motor pattern (i.e., the contribution of

each muscle in the execution of the movement), although it tends to be different from the contralateral limb.

## 5. Conclusions

The introduction of robotic systems for rehabilitation can favor a progressive recovery of distal motor ability in chronic stroke subjects in performing challenging manipulation tasks. Understanding if hand muscle patterns present alterations and whether robot-aided rehabilitation allows improving these patterns is certainly ambitious, taking into account that the CNS control strategies are not yet completely clear, especially for the hand. The preliminary analysis carried out in this work on seven post-stroke patients revealed that the muscular synergies of the healthy hand and the injured one before and after robot-mediated rehabilitation present a good level of similarity. The performed analysis shows how, despite very similar muscular patterns between the injured and the healthy limb, there is an evident reduction in terms of mobility and capacity of manipulation of the patient's hand. The involved subjects also show a partial recovery following the robot-aided rehabilitation treatment, also testified by the results of the clinical scales. However, although the present study shows an improvement in terms of the level of similitude of the muscle synergy patterns before and after the rehabilitation treatment, it will be necessary to carry out more in-depth studies involving more patients to evaluate and estimate the effectiveness of robot-mediated rehabilitation treatment. Future works will be devoted to involving a higher number of chronic stroke patients to estimate the improvement in hand muscular synergies after robot-aided rehabilitation and to extend this study to acute and chronic stroke patients during the execution of isometric tasks, in order to verify and evaluate the improvement of distal muscle activation patterns in both static and dynamic conditions.

**Author Contributions:** Conceptualization, F.S.d.L. and F.C.; Data curation, F.S.d.L., F.C., M.B., F.S. and L.Z.; Formal analysis, F.S.d.L., F.C. and L.Z.; Funding acquisition, L.Z.; Investigation, F.S.d.L., M.B., F.S. and F.B.; Methodology, F.C., M.B., F.B. and S.S.; Project administration, S.S. and L.Z.; Resources, F.S.; Software, F.S.d.L.; Supervision, F.C., F.B., S.S. and L.Z.; Validation, F.S.; Visualization, F.S.d.L. and M.B.; Writing—original draft, F.S.d.L., F.C. and L.Z. All authors have read and agreed to the published version of the manuscript.

**Funding:** This work was supported partly by Regione Lazio with HeAL9000 project (CUP: B84I20001880002), partly by the Italian Institute for Labour Accidents (INAIL) with the BioARM (CUP: E58D19000650005) and WI-FI MyoHand projects (CUP: E59E19001460005) and partly by the European Union's Horizon 2020 Research and Innovation Programme with ODIN project (CUP: C85F21000670006).

**Institutional Review Board Statement:** The study was conducted according to the guidelines of the Declaration of Helsinki, and approved by the Ethics Committee of Università Campus Bio-Medico di Roma (Ethical Approval N. 41/17 OSS Com Et CBM).

**Informed Consent Statement:** Written informed consent was obtained from all subjects involved in the study.

**Data Availability Statement:** Data and software code will be made available by materials transfer agreement upon reasonable request.

**Conflicts of Interest:** The authors declare no conflict of interest. The funders had no role in the design of the study; in the collection, analyses, or interpretation of data; in the writing of the manuscript, or in the decision to publish the results.

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
