# Peer review of "Modification of Hand Muscular Synergies in Stroke Patients after Robot-Aided Rehabilitation"

_applsci, doi:10.3390/app12063146_

Round 1

Reviewer 1 Report

This paper studied robotic systems for rehabilitation in chronic stroke patients. The use of robot-assisted training appears to be a promising method in chronic stroke subjects.  Here are the major comments,

  1. 7 subjects might affect the statistical significance.
  2. Treatment group vs. control group might shed more light on this study.
  3. The sample size calculation needs to be taken into consideration.

Author Response

Thank you.

Reviewer 2 Report

1. The authors analyzed the ability of the robot to improve limb coordination in rehabilitation training by comparing the hand EMG signals of patients before and after treatment with the exoskeleton robot, and this study is more meaningful.
2. This chapter mainly focuses on the collection and analysis of the later experiments, the samples collected are small, and the validity reflected in the limited sample data is questionable, and it is hoped that the amount of data can be increased appropriately.
3. No first line indent is taken in paragraphs 2-7 of the introduction.

Author Response

Thank you.

Reviewer 3 Report

The paper approaches an important topic on muscle synergy regarding robotic therapy for the impaired upper extremities. In rehabilitation, it is an increased need to identify protocols of robot rehabilitation and to understand how these devices shall be used for goal setting and motor function recovery. The authors should reconsider the following aspects:

In the introduction, the authors provide information regarding "alterations in the activation of muscular patterns" (line 68) on post-stroke patients. I recommend that the authors should also emphasise the CNS reorganization and neuroplasticity importance in motor function for post-stroke patients.

Materials and methods-
1. The authors should provide sample size power and size effect for their research, also discuss in limitation section regarding the small number of subjects enrolled in the research.

2. Also regarding limitations- the authors should emphasize that chronic post-stroke patients were involved in the research- a situation in which neuroplasticity occurs slower, compared to acute and especially subacute post-stroke stages.

Line 142- Please specify how was the Motor Power assessed. It is not clear.

Line 148- The authors should detail that the grasping was performed similarly with the Fugl Meyer assessment.

Lines 167-169 The authors should point out how exactly was the Gloreha used-regarding the sensors from the healthy upper extremities to replicate motions on the affected limb. ( It is not clear from the description).

After line 183- the first sentence- is NMF=NNMF?
Based on what data the synergy extraction was iterated 50 times? Does it influence the research results? Please detail.

Also in the result section- the authors did not detail which muscles performed synergies during the upper extremity motions. Otherwise, the authors refer to the synergy between healthy limb and affected limb? If so, the authors should better explain this aspect in the research aims, and in the introduction.

Author Response

Thank you.
